# Post-Diagnostic Statin Use Reduces Mortality in South Korean Patients with Dyslipidemia and Gastrointestinal Cancer

**DOI:** 10.3390/jcm10112361

**Published:** 2021-05-27

**Authors:** Kyu-Tae Han, Seungju Kim

**Affiliations:** 1Division of Cancer Control and Policy, National Cancer Control Institute, National Cancer Center, Goyang 10408, Korea; kthan.phd@gmail.com; 2Department of Nursing, College of Nursing, The Catholic University of Korea, Seoul 06591, Korea

**Keywords:** statin, medication possession ratio, gastrointestinal cancer, dyslipidemia, daily defined dose, mortality, 5-year mortality

## Abstract

Background: Statins play a role in lowering serum cholesterol and are known to have pleiotropic effects in a variety of diseases, including cancer. Despite the beneficial effects of statins in dyslipidemia patients, the treatment rate for dyslipidemia in Korea remains low, and evidence supporting the continued use of statins is lacking. The purpose of this study was to evaluate the effect of continued statin use and dosage on patient mortality after diagnosis of dyslipidemia and gastrointestinal (GI) cancer. Methods: We used data from the National Health Insurance Sampling (NHIS) cohort to evaluate patients diagnosed with dyslipidemia from 2002 to 2015. A total of 901 GI cancer patients with dyslipidemia and 62,727 non-cancer dyslipidemia patients were included in the study. During the study period, each patient’s medication possession ratio (MPR) after diagnosis was evaluated as a measure of continued statin use. Statin dosage was measured based on a defined daily dose (DDD). Finally, we used Cox-proportional hazard ratios to identify associations between the continual use of statins and mortality in patients with dyslipidemia and GI cancer. Results: In our study, mortality decreased with increasing MPR and reached significance in MPRs exceeding 50% for GI cancer patients and 75% for dyslipidemia patients compared to patients that did not use statins. Moreover, patients with high MPRs had significantly reduced 5-year mortality compared to non-users, and cause-specific mortality analyses revealed that high MPR was associated with decreased colorectal cancer death. We did not find a significant dose–response relationship between statins and mortality. Conclusion: Our findings suggest that continued statin use after diagnosis is associated with reduced patient mortality. Altogether, these results support the continued use of statins in dyslipidemia patients with and without GI cancer and highlight the importance of patient education by healthcare providers.

## 1. Background

Globally, cancer is a leading cause of death and was responsible for an estimated 9.6 million deaths in 2018 [1]. Additionally, 18.1 million patients were diagnosed with cancer in 2018, highlighting the impact of cancer across the world [1]. In Korea, cancer is also a major cause of death, and the incidence and mortality rates of cancer in 2018 were 270.4 and 268.3 per 100,000, respectively, accounting for 26.5% of all deaths [2]. Although advances in medical technology have improved cancer mortality in many countries, some cancers remain difficult to treat [3]. Appropriate management of cancer patients can improve patient outcomes, especially for patients with pre-existing chronic conditions [4,5]. As a chronic disease, dyslipidemia is one of the major risk factors for cardiovascular disease (CVD) and cancer, and proper management of dyslipidemia may affect the incidence of cancer and cancer patient outcomes [6,7].

In a 2013 study of Korean adults, the prevalence of dyslipidemia was 16.58% in middle-aged adults and gradually increased with age, peaking in women over 50 years old [8]. As the Korean population ages, the number of patients with dyslipidemia is expected to increase, but the diagnosis and treatment rates for dyslipidemia remain low [8]. According to previous studies, the treatment rate of dyslipidemia in Korea is a mere 11.9%, and approximately 50% of cancer survivors do not receive treatment for dyslipidemia [9]. Although appropriate management is considered necessary, it has not been implemented.

Statins are a class of drugs used to lower serum cholesterol in patients with dyslipidemia, and the use of statins reduces the development of secondary disease [10]. For example, statins are associated with decreased incidence of CVD [11,12], and statin use in patients with CVD risk factors improves survival rates and decreases the risk of CVD [11]. Statins have pleiotropic effects, including reducing vascular inflammation, decreasing smooth muscle proliferation, and immunomodulation [13,14,15]. In addition, using statins to achieve healthy serum cholesterol levels protects against cancer risk [16], reduces cancer-related mortality, and increases survival rates for colorectal and pancreatic cancer [17,18,19]. 

Although many studies have shown positive effects of statin use, there is insufficient information on the effect of continued statin use after diagnosis of dyslipidemia in Asian populations. In addition, although the mechanism of association between serum cholesterol and cancer incidence is unclear, it has been suggested that abnormal serum cholesterol levels influence the development of gastrointestinal (GI) cancer, including esophageal and colon cancer [20]. Given these findings, further research is needed on the effect of the continued use of statins on patients’ outcomes, especially patients with dyslipidemia diagnosed with gastrointestinal cancer. 

Appropriate management through continued use of statins may lead to better results in patients [21]; however, statin use may have different effects in patients with dyslipidemia and dyslipidemia patients with cancer. In this study, we evaluated the effects of continued use of post-diagnostic statins and the doses of these drugs on mortality in dyslipidemia patients with or without GI cancer. The use of statins in dyslipidemia patients was evaluated after initial diagnosis, and GI cancer patients were evaluated for continued statin use after cancer diagnosis. Subgroup analysis was conducted by sex for the effects of continued statin use on patient mortality. Our findings highlight the importance of actively monitoring cholesterol management in dyslipidemia and GI cancer patients, and have implications for clinicians, especially in Korea.

## 2. Methods

### 2.1. Database and Data Collection

This study used data collected from the National Health Insurance Sampling (NHIS) cohort between 2002 and 2015. The cohort consisted of a baseline population of 1,025,340 randomly selected participants and represented 2.2% of the total eligible Korean population in 2002 [22]. The data included demographic information, treatment data, date of death, and hospital characteristics. Medical data for all subjects were available as part of the insurance claim and included diagnosis, comorbidities, medications, date of visit, and cost.

During the study period, 124,218 patients were diagnosed with dyslipidemia, which was defined according to the International Classification of Diseases (ICD-10 code: E78). We included patients who were first diagnosed with dyslipidemia to reduce time-related bias, and excluded patients whose death was less than 6 months after diagnosis [23]. First, we excluded patients diagnosed with dyslipidemia or who were prescribed statins between 2002 and 2003 to include patients diagnosed with dyslipidemia for the first time during the study period. Next, we excluded patients with other diseases, such as diabetes, which can affect cancer incidence and death [24]. Third, we excluded patients under 30 and over 75 years of age. Next, we excluded patients with missing variables, such as body mass index (BMI). We also excluded patients diagnosed with any carcinoma other than GI cancer to reduce heterogeneity due to carcinoma. GI cancer was defined based on the ICD 10 code as follows: gastroesophageal (C15, C16), colorectal (C18, C19, C20, C21), and hepatobiliary pancreatic cancer (C22, C23, C24, C25). For inclusion of GI cancer patients, we evaluated the date of cancer diagnosis. GI cancer patients who were diagnosed with cancer before dyslipidemia were excluded. Additionally, we excluded patients who died before the diagnosis of dyslipidemia or within 6 months of diagnosis. Similarly, we excluded GI cancer patients who died before the diagnosis of GI cancer or within 6 months after the GI cancer diagnosis. Ultimately, 63,628 patients with dyslipidemia and GI cancer were included in the study. 

### 2.2. Variables

In this study, we considered variables related to statin use, mortality, and patient demographics as follows. As we were interested in the continuous use of statins, we first calculated the medication possession ratio (MPR) for each patient during the study period [25]. We included medications from the diagnosis of dyslipidemia to the date of death or the end of the study. In patients with GI cancer, there may be biases related to statin treatment time, and to reduce this, we considered taking statins in the initial diagnosis of dyslipidemia in the same way as patients with non-cancerous dyslipidemia [23]. First, we included all statin medications prescribed starting on the date of initial diagnosis in patients with dyslipidemia and until the end of the study or death. We calculated the sum of the total prescription days per patient based on the drugs prescribed each year of the study period. Next, we modified the total supply days to not exceed 365 days. Second, the average prescription days for the study period were obtained by dividing the total prescription days of each patient each year by the patient’s observation period (years). Third, the total prescribed days were divided by 365 to calculate the annual MPR [25].
$$\mathrm{MPR}=\frac{All\;days\;supply\left(\leq 365\right)}{365}$$

Finally, we classified MPR into quartile categories: ≤25%, ≤50%, ≤75%, and >75%. All MPRs are average prescriptions for the study periods, and those who were not prescribed statins at least once were classified as non-users. In addition, we calculated the statin dose using the total supply in days and the quantity of statins prescribed, which included simvastatin, lovastatin, pravastatin, fluvastatin, atorvastatin, and pitavastatin. The cumulative definition daily dose (DDD) was calculated by calculating the patient’s statin prescription each year in the same way as the MPR and dividing the total number of observations based on the actual prescription year.

We evaluated patient death after dyslipidemia or GI cancer diagnosis as a measure of patient outcomes. We considered all-cause mortality in patients with dyslipidemia during the study period. Patients with dyslipidemia were evaluated from dyslipidemia diagnosis to death or the end of the study. Similarly, GI cancer patients were evaluated from cancer diagnosis to death or the end of the study. In patients with GI cancer, the type of cancer was included as a variable (gastroesophageal cancer, colorectal cancer, or hepatobiliary pancreatic cancer). The continuous management of dyslipidemia can be affected by the treatment hospital. Therefore, the hospital where the patient was primarily treated was also considered as a variable. First, we calculated the number of visits to all medical institutions where patients were prescribed statins for dyslipidemia during the study period. We then determined the most visited type of hospital by dividing the number of visits to each institution by the total number of visits. For patients who were not prescribed statins, the hospital of diagnosis was considered the main medical institution. Finally, medical institutions were classified into community health centers and clinics, hospitals, general hospitals, and tertiary hospitals. 

Patient demographic data included sex (male or female) and age (30–44, 45–59, or 60–75). We classified residence areas based on the 17 administrative districts in Korea: Seoul and Gyeonggi-do were classified as the capital area, 7 metropolitan areas were classified as the metropolitan area, and the rest were classified as the other area. Income was measured based on NHI insurance premiums, which pay a regular portion of their average salary or property. Depending on the level of premium paid by the individual, it was classified into ‘low’ (~30th percentile), ‘low-moderate’ (31st to 60th percentile), ‘moderate-high’ (61st to 80th percentile), and ‘high’ (81th percentile and above) groups. The average body mass index (BMI) was measured and classified into five categories based on the Asian population: under-weight (<18.5 kg/m^2^), normal (18.5–22.9 kg/m^2^), overweight (23–24.9 kg/m^2^), obese I (25–29.9 kg/m^2^), and obese II (≥30 kg/m^2^) [26]. The patient’s severity was measured by the Charlson Comorbidity Index (CCI), and diabetes was not reflected in the score because all patients with diabetes were excluded [27]. The year of dyslipidemia diagnosis (2004–2007, 2008–2011, or 2012–2015) was included.

### 2.3. Ethical Consideration

This study was approved by the Institutional Review Board of Eulji University (IRB number: EUIRB2020-025).

### 2.4. Statistical Analysis

The distribution of each categorical variable was examined by analyzing frequencies and percentages with χ^2^ tests. For continuous variables, t-tests were performed to compare mean and standard deviation values. We used the Cox-proportional hazard ratio (HR) to identify the association between continuous use of statins and mortality in dyslipidemia patients. For this analysis, the start date was defined as the date of diagnosis of dyslipidemia or GI cancer, and the end date was defined as the date of death or the end of study (31 December 2015). All variables were entered simultaneously into a fully adjusted model. We evaluated the 5-year mortality rate of GI cancer patients to assess long-term effects. For this analysis, the start date was defined as the date of cancer diagnosis, and the end date was defined as 5 years from the date of diagnosis or the study end date. Cause-specific mortality was evaluated in patients diagnosed with colorectal cancer. Finally, we conducted subgroup analyses by sex to evaluate associations between the continual use of statins and mortality in dyslipidemia and GI cancer patients. All statistical analyses were performed using SAS statistical software version 9.4 (SAS Institute, Cary, NC, USA). A *p*-value <0.05 was considered statistically significant. 

## 3. Results

This study considered 62,727 non-cancer patients with dyslipidemia and 901 dyslipidemia patients with GI cancer (Table 1). GI cancer included primarily gastroesophageal (*n* = 342, 38.0%), colorectal (*n* = 341, 37.8%), and hepatobiliary pancreatic cancer (*n* = 218, 24.2%). During the study period, 458 dyslipidemia patients (0.7%) and 130 GI cancer patients (14.4%) died. In non-cancer dyslipidemia patients, we did not find any significant differences in DDD/365 between patients who died and those who survived (*p* = 0.1763). In contrast, the mean DDD/365 of surviving GI cancer patients (M = 0.61) was significantly higher than those who died (M = 0.53, *p* = 0.0424). The average DDD/365 for different statin types is shown in Appendix A Appendix A (see Appendix A). Based on the statin MPRs after GI cancer diagnosis, patients who did not use statins had higher mortality (*n* = 36, 19.6%), and mortality rates decreased significantly as MPR increased (≤25%: 14.3%; ≤50%: 18.2%; ≤75%: 9.9%; >75%: 7.6%, *p* = 0.0073). Similarly, significant differences in MPR were found between those who died and those who survived in non-cancer patients with dyslipidemia (*p* = 0.0039). The median patient survival period (months) was 29.0 for GI cancer patients and 55.0 for non-cancer patients. Appendix A shows the MPR according to the patient’s general characteristics. MPRs exceeding 75% in cancer patients and dyslipidemia were 13.1% and 9.5%, respectively (see Appendix A).

To further evaluate the relationship between the continued use of post-diagnostic statins and all-cause mortality, we performed a multivariate Cox regression analysis (Table 2). In GI cancer patients, mortality decreased as DDD/365 increased (HR: 0.910, 95% confidence interval (CI): 0.508–1.603) but was not statistically significant. Compared to GI cancer patients who did not use statins (non-users), the risk of mortality of GI cancer patients gradually decreased as MPR increased but was significant only when MPR exceeded 50% (≤75% HR: 0.373, 95% CI: 0.178–0.781; >75% HR: 0.293, 95% CI: 0.121–0.710). In non-cancer patients with dyslipidemia, mortality decreased as DDD/365 increased (HR: 0.820, 95% CI: 0.566–1.186) but was not statistically significant. Patients with higher MPR (>75%: HR: 0.550, 95% CI: 0.349–0.867) had a significantly reduced risk of mortality compared to non-users. 

To further define the suggested beneficial impact of continued statin use on patient outcomes, we evaluated 5-year mortality in patients with GI cancer as well as cause-specific mortality in patients with colorectal cancer (Figure 1). The 5-year mortality significantly decreased with increasing MPR compared to non-users (≤75% HR: 0.362, 95% CI: 0.177–0.742; >75% HR: 0.258, 95% CI: 0.109–0.613) but was not associated with DDD/365. Analysis of cause-specific mortality revealed that high MPR was associated with decreased mortality of colorectal cancer patients compared to non-users, but it was statistically significant when MPR exceeded 75% (HR: 0.164, 95% CI: 0.030–0.888).

Finally, we performed subgroup analyses of continual statin use and GI cancer by sex (Table 3). For patients with GI cancer, an increase in DDD/365 was associated with a decrease in mortality in males and an increase in mortality in females but was not statistically significant. Overall high MPR in male and female GI cancer patients was associated with reduced mortality, but was statistically significant only when MPR was less than 75% in females (HR: 0.288, 95% CI: 0.094–0.884). For non-cancer patients with dyslipidemia, high DDD/365 was associated with decreased mortality in both males and females but was not statistically significant. In female dyslipidemia patients, low MPR was associated with increased mortality (≤25% HR: 2.753, 95% CI: 1.429–5.307). 

## 4. Discussion

Statins lower serum cholesterol levels and can affect diseases unrelated to cholesterol [14]. Although continued statin use can affect a patients’ secondary diseases and is associated with better outcomes, the treatment rate for dyslipidemia in Korea remains low. In this study, we found that high MPR after diagnosis was associated with reduced mortality in dyslipidemia patients with and without GI cancer. We did not find a dose–response relationship between statin use and mortality. Similar results have been suggested in previous studies. For example, the use of statins after diagnosis in ovarian and prostate cancer is associated with improved patient survival [28,29]. Use of simvastatin and atorvastatin after pancreatic cancer diagnosis is also associated with longer survival rates. Furthermore, a dose–response relationship between cumulative DDD and reduced mortality has been reported in prostate cancer patients [30]. In a Japanese study, similar mortality rates were observed in patients receiving statin monotherapy and lifestyle modifications, which were both associated with reduced deaths in colorectal cancer but were not statistically significant [31]. Together with our present findings, these studies highlight the benefit of continued statin use after cancer diagnosis. Our results suggest that continuity of statin use is important in both dyslipidemia and GI cancer patients and non-cancer patients. 

The beneficial effects of continued statin use on patient mortality after diagnosis may be related to the pleiotropic effects of statins. Cholesterol is a major structure in cells, and the pathway for synthesizing cholesterol is similar to that of cells [32]. Statins affect the cardiovascular system by inhibiting HMG-CoA reductase, reducing serum cholesterol levels, preventing vascular smooth muscle thickening, and reducing cell proliferation [13]. In addition, statins inhibit tumor cell growth and angiogenesis, induce apoptosis, and positively affect the treatment of cancer patients [32]. Previous studies have suggested a synergistic interaction between statins and chemotherapy, and statins may play a role in enhancing the antitumor activity of various cytokines [33,34]. Recent studies have shown that using statins with an immune checkpoint inhibitor (ICI) has been associated with better patient outcomes in some carcinomas [35,36]. In these studies, high-intensity statin use was associated with better clinical outcomes, including the objective response rate and progression-free survival [36]. 

Many studies have shown that statins can lead to better clinical outcomes in patients, and our findings have suggested the importance of continuity of statin use in patients with dyslipidemia. Indeed, our study found that statin MPRs exceeding 50% in dyslipidemia patients with GI cancer were associated with significantly reduced mortality compared to statin non-users. Patients with dyslipidemia showed different trends, and when MPR was maintained above 75%, it was associated with a significant reduction in mortality. These results suggest that patients with dyslipidemia should be aware of the need for continued use of statins. Our results highlight the importance of healthcare providers communicating the benefit of treatment to patients with dyslipidemia. To increase treatment rates for patients with dyslipidemia, healthcare providers need to emphasize the importance of continued statin use and maintain therapeutic relationships to ultimately improve patient outcomes.

Although we found that increased statin dose (DDD/365) was associated with decreased mortality in GI cancer patients and non-cancer patients with dyslipidemia, these observations were not statistically significant. Previously, a dose–response relationship was identified between increased DDD and reduced all-cause mortality and cancer-specific mortality [37]. In contrast, no clinical benefit was observed with the addition of a low dose of simvastatin to chemotherapy in pancreatic cancer patients [38]. A Danish study also found no dose–response relationship between statin use and risk of death after prostate cancer diagnosis [29]. Therefore, the relationship between the dose response and cancer mortality requires additional studies that consider the effect of specific statins and cancer types.

Our subgroup analysis showed that the effects of post-diagnosis statin use on GI cancer mortality varied with gender. In female dyslipidemia patients, lower MPR corresponded with increased mortality compared to non-users. Previous studies have shown that females are more likely than males to stop taking statins [39]. Our results imply that continued use of statins is particularly important for females and suggest that communication between patients and healthcare providers should emphasize the positive consequences of continued statin use.

As this study included a representative population, our findings are meaningful to the broader Korean population. Importantly, our evaluation supports the continued use of statins after diagnosis in dyslipidemia patients with and without GI cancer. Despite this strength, our study has some limitations. First, we did not consider detailed treatment modalities, such as chemotherapy, after diagnosis of cancer, and some cancer treatments may affect patient outcomes. To reduce such effects, we included only GI cancer patients, and we excluded patients who died within 6 months of diagnosis. Second, the effect of statins on patient outcomes may vary depending on the type of statin prescribed. Here, we did not consider specific statin types. Therefore, further research is necessary to clarify the effects of specific statins on GI cancer patient outcomes. Third, we analyzed all-cause mortality and we realize that specific causes may affect the study results. To address this concern, we analyzed cause-specific mortality in colorectal cancer, and our results were similar to our analysis of all-cause mortality. In addition, our data do not include the stage of cancer, and patient outcomes may vary depending on the stage. Finally, individual patient factors that could not be measured in this study may affect patient outcomes, and further studies are needed to consider these factors.

## 5. Conclusions

Our findings suggest that sustained statin use after diagnosis in non-cancer patients with dyslipidemia and in GI cancer patients with dyslipidemia is associated with reduced mortality compared to those that did not take statins. These results have important implications for healthcare providers. Specifically, healthcare providers need to emphasize the importance of continued statin use to patients with dyslipidemia and to female patients in particular. Our findings also support the education of patients with dyslipidemia regarding continued statin use to improve treatment rates. Altogether, this study serves as an important validation of the benefit of continued statin use in dyslipidemia patients and extends this utility to dyslipidemia patients with GI cancer.

## Figures and Tables

**Figure 1 jcm-10-02361-f001:**
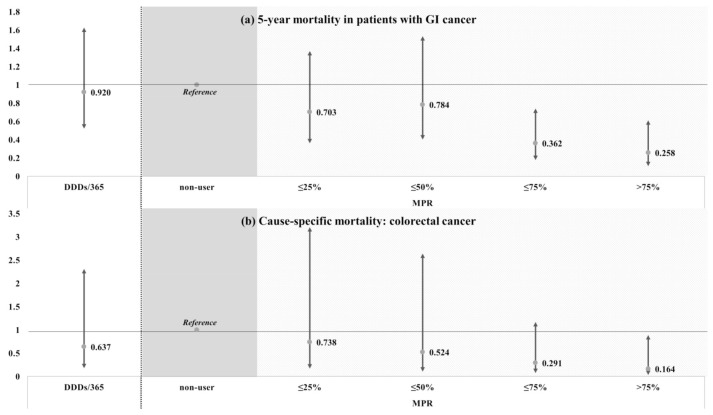
Effect of continued statin use on mortality in GI cancer patients. (**a**) Relationship between continued use of statin and 5-year mortality. (**b**) Relationship between continued use of statins and death from colorectal cancer.

**Table 1 jcm-10-02361-t001:** Association between continued statin use and mortality after diagnosis in dyslipidemia and GI cancer patients (Unit: N/M, % /SD).

	GI Cancer Patients				*p*-Value	Survival Time (Months)			Non-Cancer Patients				*p*-Value	Survival Time (Months)		
	Died		Survival			Median	95% CI		Died		Survival			Median	95% CI	
**DDDs/365**	0.53	±0.44	0.61	±0.46	0.0424				0.58	±0.41	0.55	±0.46	0.1763			
**MPR**																
Non-user	36	(19.6)	148	(80.4)	0.0073	24.0	21.0	28.0	96	(0.6)	16,796	(99.4)	0.0039	46.0	46.0	47.0
≤25%	39	(14.3)	233	(85.7)		28.0	20.0	34.0	136	(0.7)	19,348	(99.3)		50.0	49.0	51.0
≤50%	30	(18.2)	135	(81.8)		29.0	24.0	37.0	90	(0.8)	11,009	(99.2)		60.0	59.0	61.0
≤75%	16	(9.9)	146	(90.1)		30.0	25.0	36.0	91	(1.0)	9,197	(99.0)		66.0	65.0	67.0
>75%	9	(7.6)	109	(92.4)		37.0	31.0	45.0	45	(0.8)	5,919	(99.2)		66.0	65.0	68.0
**Cancer types**																
Gastroesophageal cancer	41	(12.0)	301	(88.0)	<0.0001	31.0	26.0	35.0								
Colorectal cancer	38	(11.1)	303	(88.9)		31.0	28.0	36.0								
Hepatobiliary pancreatic cancer	51	(23.4)	167	(76.6)		23.0	19.0	27.0								
**Main treatment medical institutions for dyslipidemia**																
Community health center and clinic	87	(14.5)	514	(85.5)	0.8755	33.0	15.0	40.0	268	(0.6)	44,165	(99.4)	<0.0001	67.0	64.0	69.0
hospital	13	(14.8)	75	(85.2)		26.0	24.0	30.0	48	(0.8)	6257	(99.2)		54.0	53.0	54.0
General hospital	23	(15.1)	129	(84.9)		34.5	28.0	41.0	99	(1.1)	8846	(98.9)		46.0	45.0	48.0
Tertiary hospital	7	(11.7)	53	(88.3)		35.5	25.0	43.0	43	(1.4)	3001	(98.6)		58.0	57.0	60.0
**CCI**	3.64	±2.44	3.22	±1.82	0.0213	31.0	22.0	39.0	3.52	±2.01	2.57	±1.67	<0.0001	71.0	69.0	74.0
**Sex**																
Male	78	(16.1)	407	(83.9)	0.2524	27.0	25.0	32.0	261	(0.9)	27,516	(99.1)	<0.0001	54.0	53.0	55.0
Female	52	(12.5)	364	(87.5)		30.0	26.0	34.0	197	(0.6)	34,753	(99.4)		56.0	55.0	56.0
**Age**																
30–44	8	(13.1)	53	(86.9)	0.0012	28.0	20.0	56.0	41	(0.3)	14,127	(99.7)	<0.0001	59.0	58.0	60.0
45–59	38	(9.9)	346	(90.1)		29.0	25.0	35.0	124	(0.4)	31,575	(99.6)		54.0	54.0	55.0
60–75	84	(18.4)	372	(81.6)		29.0	25.0	32.0	293	(1.7)	16,567	(98.3)		53.0	52.0	54.0
**BMI**																
<18.5	1	(14.3)	6	(85.7)	0.1326	72.0	4.0	105.0	14	(1.6)	837	(98.4)	0.0180	46.0	43.0	49.0
18.5–22.9	48	(17.6)	225	(82.4)		29.0	25.0	34.0	156	(0.8)	19,987	(99.2)		51.0	50.0	52.0
23–24.9	27	(10.0)	242	(90.0)		29.0	24.0	34.0	115	(0.7)	17,461	(99.3)		55.0	54.0	56.0
25–29.9	51	(15.7)	274	(84.3)		27.0	24.0	33.0	157	(0.7)	21,614	(99.3)		58.0	57.0	59.0
≥30	3	(17.6)	14	(82.4)		37.0	19.0	58.0	16	(0.7)	2,370	(99.3)		58.0	56.0	60.0
**Residence area**																
Capital area	69	(16.8)	342	(83.2)	0.1588	32.0	26.0	37.0	182	(0.7)	26,996	(99.3)	0.1507	56.0	56.0	57.0
Metropolitan	25	(11.5)	192	(88.5)		29.0	25.0	35.0	123	(0.7)	16,926	(99.3)		54.0	53.0	55.0
Other	36	(13.2)	237	(86.8)		26.0	22.0	30.0	153	(0.8)	18,347	(99.2)		54.0	53.0	55.0
**Income**																
Low	28	(14.5)	165	(85.5)	0.9995	29.0	23.0	36.0	127	(0.9)	14,400	(99.1)	0.0143	52.0	51.0	53.0
Low-moderate	30	(14.6)	175	(85.4)		28.0	24.0	33.0	123	(0.8)	15,139	(99.2)		54.0	53.0	55.0
Moderate-high	30	(14.2)	181	(85.8)		28.0	24.0	34.0	95	(0.7)	13,883	(99.3)		55.0	54.0	56.0
High	42	(14.4)	250	(85.6)		29.5	25.0	35.0	113	(0.6)	18,847	(99.4)		58.0	57.0	59.0
**Year of diagnosis**																
2004–2007	75	(19.8)	303	(80.2)	0.0002	45.0	39.0	54.0	231	(1.9)	11,956	(98.1)	<0.0001	118.0	117.0	118.0
2008–2011	43	(11.7)	325	(88.3)		27.5	25.0	32.0	177	(0.8)	22,563	(99.2)		70.0	70.0	71.0
2012–2015	12	(7.7)	143	(92.3)		14.0	12.0	17.0	50	(0.2)	27,750	(99.8)		24.0	23.0	24.0
**Total**	130	(14.4)	771	(85.6)		29.0	26.0	32.0	458	(0.7)	62,269	(99.3)		55.0	54.0	55.0

BMI, body mass index; CCI, Charlson comorbidity index; DDD, defined daily dose; GI, gastrointestinal; MPR, medication possession ratio; M, nean; SD, standard deviation; 95% CI, 95% confidence interval.

**Table 2 jcm-10-02361-t002:** The multivariate Cox regression analysis of the association of continued statin use and mortality after diagnosis in dyslipidemia patients.

		GI Cancer Patients				Non-Cancer Patients			
		HR	95% CI		*p*-Value	HR	95% CI		*p*-Value
**DDD/365**		0.910	0.508	1.603	0.7510	0.820	0.566	1.186	0.2914
**MPR**									
Non-user		Ref	-	-		Ref	-	-	
≤25%		0.716	0.380	1.349	0.3011	1.337	0.916	1.952	0.1326
≤50%		0.821	0.419	1.608	0.5654	1.138	0.761	1.702	0.5289
≤75%		0.373	0.178	0.781	0.0089	1.001	0.669	1.499	0.9953
>75%		0.293	0.121	0.710	0.0066	0.550	0.349	0.867	0.0100
**Cancer types**									
Gastroesophageal cancer		0.464	0.302	0.714	0.0005				
Colorectal cancer		0.464	0.297	0.723	0.0007				
Hepatobiliary pancreatic cancer		Ref	-	-					
**Main treatment medical institutions for dyslipidemia**									
Community health center and clinic		Ref	-	-		Ref	-	-	
hospital		0.821	0.452	1.492	0.5175	1.456	1.069	1.983	0.0173
General hospital		0.969	0.602	1.559	0.8952	1.741	1.376	2.203	<0.0001
Tertiary hospital		0.891	0.399	1.992	0.7792	2.028	1.451	2.834	<0.0001
**CCI**		1.124	1.013	1.220	0.0253	1.282	1.219	1.348	<0.0001
**Sex**									
Male		1.402	0.969	2.030	0.073	2.283	1.887	2.762	<0.0001
Female		Ref	-	-		Ref	-	-	
**Age**									
30–44		Ref	-	-		Ref	-	-	
45–59		0.919	0.413	2.046	0.8363	1.410	0.980	2.029	0.0644
60–75		1.761	0.776	3.994	0.1759	4.908	3.401	7.081	<0.0001
**BMI**									
<18.5		0.904	0.117	6.960	0.9229	2.191	1.265	3.793	0.0051
18.5–22.9		Ref	-	-		Ref	-	-	
23–24.9		0.531	0.327	0.862	0.0104	0.713	0.560	0.908	0.0061
25–29.9		0.862	0.574	1.293	0.4717	0.778	0.623	0.972	0.0274
≥30		0.504	0.152	1.665	0.2607	0.913	0.544	1.534	0.7317
**Residence area**									
Capital area		1.513	1.000	2.301	0.0526	0.895	0.720	1.112	0.3181
Metropolitan		0.906	0.569	1.642	0.9006	0.975	0.768	1.237	0.8349
Other		Ref	-	-		Ref	-	-	
**Income**									
Low		1.257	0.766	2.063	0.3654	1.759	1.362	2.271	<0.0001
Low-moderate		1.173	0.726	1.895	0.5143	1.570	1.213	2.030	0.0006
Moderate-high		1.276	0.789	2.063	0.3203	1.244	0.095	1.637	0.1185
High		Ref	-	-		Ref	-	-	
**Year of diagnosis**									
2004–2007		Ref	-	-		Ref	-	-	
2008–2011		0.655	0.418	1.027	0.0651	0.532	0.415	0.682	<0.0001
2012–2015		0.663	0.335	1.312	0.2382	0.602	0.416	0.872	0.0072

95% CI, 95% confidence interval; BMI, body mass index; CCI, Charlson comorbidity index; DDD, defined daily dose; GI, gastrointestinal; HR, hazard ratio; MPR, medication possession ratio.

**Table 3 jcm-10-02361-t003:** Association between continued post-diagnostic statin use and mortality by gender.

	Male				Female			
	HR	95% CI		*p*-Value	HR	95% CI		*p*-Value
**GI Cancer Patients**								
**DDD/365**	0.452	0.129	1.585	0.2150	1.879	0.876	4.030	0.1054
**MPR**								
Non-user	Ref	-	-		Ref	-	-	
<25%	1.214	0.434	3.398	0.7113	0.402	0.145	1.111	0.0790
<50%	1.341	0.444	4.046	0.6026	0.454	0.145	1.421	0.1750
<75%	0.655	0.194	2.217	0.4966	0.288	0.094	0.884	0.0295
≥75%	0.206	0.035	1.196	0.0783	0.327	0.098	1.088	0.0683
**Non-Cancer Patients**								
**DDD/365**	0.980	0.636	1.511	0.9277	0.581	0.307	1.099	0.0950
**MPR**								
Non-user	Ref	-	-		Ref	-	-	
≤25%	0.906	0.570	1.441	0.6773	2.753	1.429	5.307	0.0025
≤50%	0.886	0.536	1.464	0.6366	1.929	0.975	3.816	0.0591
≤75%	0.814	0.491	1.351	0.4262	1.631	0.826	3.221	0.1587
>75%	0.472	0.267	0.834	0.0097	0.805	0.377	1.718	0.5740

95% CI, 95% confidence interval; DDD, defined daily dose; GI, gastrointestinal; HR, hazard ratio; MPR, medication possession ratio.

## Data Availability

Data was obtained from NHIS and available with the permission of NHIS.

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
