# Peer review of "Post-Diagnostic Statin Use Reduces Mortality in South Korean Patients with Dyslipidemia and Gastrointestinal Cancer"

_jcm, 2021, doi:10.3390/jcm10112361_

Round 1

Reviewer 1 Report

The authors have improved their work, but there are still some unclear points to be improved.

  • At line 103 the sentence “we excluded GI cancer patients who died before diagnosis of GI cancer… “ is still incorrect.

  • In the “Results” section the results of univariate and multivariate analysis are not clearly showed. You should specify both in the text and in the table 2 (for example writing on the table headline “univariate analysis” and “multivariate analysis”) the results of the univariate and multivariate analysis to make more clearly your results.

  • In the “Results” section you mentioned cause-specific mortality only for colon cancer patients and not for the others GI cancer patients. Please could you explain this choice?

  • Could you better define medication possession ratio (MPR)? It could be inaccurate as a value since the various drugs could have packages with different number of tablets inside. Could you insert a reference for it? Maybe this work could help you “Tang KL, Quan H, Rabi DM. Measuring medication adherence in patients with incident hypertension: a retrospective cohort study. BMC Health Serv Res. 2017 Feb 13;17(1):135. doi: 10.1186/s12913-017-2073-y. PMID: 28193217; PMCID: PMC5307770.”

  • Taking into account table 1. Instead of the mean and standard deviation it is recommended to insert the median and the confidence interval (CI), calculated with Kaplan meier.

  • In the “Results” section p values were not showed, please add them.

  • “Results” section: “For patients with GI cancer, an increase in DDD/365 was associated with a decrease in mortality in male and an increase in mortality in female, but was not statistically significant”. For this kind of subgroup analysis you should calculate the p-value for  interaction test.

Author Response

First, we greatly appreciate the comments and suggestions offered by the reviewers, which we used to improve the manuscript. Our response to each comment follows, and we have attached a revision note and also highlighted the revised sections of the manuscript. Again, thank you for the valuable and helpful comments.

Answer to Reviewer 1:

The authors have improved their work, but there are still some unclear points to be improved. 

At line 103 the sentence “we excluded GI cancer patients who died before diagnosis of GI cancer… “ is still incorrect.

 Answer: We apologize for the lack of information on the study population.

In this study, patients who died before or within 6 months after diagnosis of dyslipidemia or GI cancer were excluded.

We have revised the sentence as follows(page 3 line 102-104).

Similarly, patients with GI cancer who died before being diagnosed with GI cancer or within 6 months after being diagnosed with GI cancer were excluded.

In the “Results” section the results of univariate and multivariate analysis are not clearly showed. You should specify both in the text and in the table 2 (for example writing on the table headline “univariate analysis” and “multivariate analysis”) the results of the univariate and multivariate analysis to make more clearly your results.

  Answer: Thank you for your comments. We revised Table 2 headline and results as following:

Table2. The multivariate Cox regression analysis of the association of continued statin use and mortality after diagnosis in dyslipidemia patients

To further evaluate the relationship between continued use of post-diagnostic statins and all-cause mortality, we performed multivariate Cox regression analysis(page 5 line 191-192).

In the “Results” section you mentioned cause-specific mortality only for colon cancer patients and not for the others GI cancer patients. Please could you explain this choice?

  Answer: Thank you for your comments. We wanted to conduct an analysis by cause of death, but the number of patients who died from the specific-cancer was insufficient. Accordingly, only patients who died from colorectal cancer were analyzed.

Could you better define medication possession ratio (MPR)? It could be inaccurate as a value since the various drugs could have packages with different number of tablets inside. Could you insert a reference for it? Maybe this work could help you “Tang KL, Quan H, Rabi DM. Measuring medication adherence in patients with incident hypertension: a retrospective cohort study. BMC Health Serv Res. 2017 Feb 13;17(1):135. doi: 10.1186/s12913-017-2073-y. PMID: 28193217; PMCID: PMC5307770.”

Answer: Thank you for your comments.

The method of calculating MPR in our study already has a reference.

<references>

  1. Kozma, C.M.; Dickson, M.; Phillips, A.L.; Meletiche, D.M. Medication possession ratio: implications of using fixed and variable observation periods in assessing adherence with disease-modifying drugs in patients with multiple sclerosis. Patient preference and adherence 2013, 7, 509.

In addition, we have added the formula below to help the authors understand (page 3 line 121).

.

Taking into account table 1. Instead of the mean and standard deviation it is recommended to insert the median and the confidence interval (CI), calculated with Kaplan meier.

Answer: Thank you for your comments. We revised Table 1 as following:

Also, in the process of modifying the table, we found that we did not modify the variable name (year of diagnosis) that was modified in revision 1, and we corrected it.

In the “Results” section p values were not showed, please add them.

  Answer: Thank you for your comments. We added the p-value to the result as following(page 4 line 177-187):

“Results” section: “For patients with GI cancer, an increase in DDD/365 was associated with a decrease in mortality in male and an increase in mortality in female, but was not statistically significant”. For this kind of subgroup analysis you should calculate the p-value for  interaction test.

 Answer: Thank you for your comments.

We evaluated the correlation between sex, DDD, and MPR to confirm the effect modifier, and the results are as follows. Accordingly, we conducted a subgroup analysis by gender.

Reviewer 2 Report

answers are satisfactory

Author Response

We greatly appreciate the comments and suggestions offered by the reviewers, which we used to improve the manuscript.

This manuscript is a resubmission of an earlier submission. The following is a list of the peer review reports and author responses from that submission.

Round 1

Reviewer 1 Report

The authors presented a retrospective study evaluating the effect of continued statin use and dosage on patient mortality in a wide cohort taken from Korean NHIS of patients affected by dyslipidemia and dyslipidemia plus GI cancers. Although their work has been really appreciable, I think that several issues should be addressed in the paper:

Background:  

1- The researchers wrote “Cancer is a leading cause of death and was responsible for an estimated 8.2 million 35 deaths in 2013”,  please insert more updated data.

2- Please explain more clearly why you chose GI cancer patients. Contextualize this choice more.

3- Researchers says that management of dyslipidemia may affect the incidence of cancer and cancer patient outcome. Add some references here. Hereby find an example:

  • Giampieri R, Cantini L, Giglio E, Bittoni A, Lanese A, Crocetti S, Pecci F, Copparoni C, Meletani T, Lenci E, Lupi A, Baleani MG, Berardi R. Impact of Polypharmacy for Chronic Ailments in Colon Cancer Patients: A Review Focused on Drug Repurposing. Cancers (Basel). 2020 Sep 23;12(10):2724. doi: 10.3390/cancers12102724. PMID: 32977434; PMCID: PMC7598185.

4- Moreover, taking into account the results of KEYNOTE-177 (cite), you should briefly introduce the emerging role of immune checkpoint inhibitors (ICIs) in GI cancer and the potential synergism between statins and ICIs. You can add this consideration in the discussion section due to space limitation in the background section.

You might take a cue from there articles:

  • André T, Shiu KK, Kim TW, Jensen BV, Jensen LH, Punt C, Smith D, Garcia-Carbonero R, Benavides M, Gibbs P, de la Fouchardiere C, Rivera F, Elez E, Bendell J, Le DT, Yoshino T, Van Cutsem E, Yang P, Farooqui MZH, Marinello P, Diaz LA Jr; KEYNOTE-177 Investigators. Pembrolizumab in Microsatellite-Instability-High Advanced Colorectal Cancer. N Engl J Med. 2020 Dec 3;383(23):2207-2218. doi: 10.1056/NEJMoa2017699. PMID: 33264544.
  • Xia Y, Xie Y, Yu Z, Xiao H, Jiang G, Zhou X, Yang Y, Li X, Zhao M, Li L, Zheng M, Han S, Zong Z, Meng X, Deng H, Ye H, Fa Y, Wu H, Oldfield E, Hu X, Liu W, Shi Y, Zhang Y. The Mevalonate Pathway Is a Druggable Target for Vaccine Adjuvant Discovery. 2018 Nov 1;175(4):1059-1073.e21. doi: 10.1016/j.cell.2018.08.070. Epub 2018 Sep 27. PMID: 30270039.
  • Cantini L, Pecci F, Hurkmans DP, Belderbos RA, Lanese A, Copparoni C, Aerts S, Cornelissen R, Dumoulin DW, Fiordoliva I, Rinaldi S, Aerts JGJV, Berardi R. High-intensity statins are associated with improved clinical activity of PD-1 inhibitors in malignant pleural mesothelioma and advanced non-small cell lung cancer patients. Eur J Cancer. 2021 Feb;144:41-48. doi: 10.1016/j.ejca.2020.10.031. Epub 2020 Dec 14. PMID: 33326868.
  • Omori M, Okuma Y, Hakozaki T, Hosomi Y. Statins improve survival in patients previously treated with nivolumab for advanced non-small cell lung cancer: An observational study. Mol Clin Oncol. 2019 Jan;10(1):137-143. doi: 10.3892/mco.2018.1765. Epub 2018 Nov 13. PMID: 30655989; PMCID: PMC6313973.

Methods:

5-Database and data collection: in this section researchers said that in this study they excluded patients with diseases, such as diabetes, because can affect cancer incidence and death. In the following section “variables”, they took into account the Charlson Comorbility Index (CCI) (please add a citation for it, for example Charlson ME, Pompei P, Ales KL, MacKenzie CR. A new method of classifying prognostic comorbidity in longitudinal studies: development and validation. J Chronic Dis. 1987;40(5):373-83. doi: 10.1016/0021-9681(87)90171-8. PMID: 3558716; Charlson M, Szatrowski TP, Peterson J, Gold J. Validation of a combined comorbidity index. J Clin Epidemiol. 1994 Nov;47(11):1245-51. doi: 10.1016/0895-4356(94)90129-5. PMID: 7722560), an index that include diabetes among its variables.

Could you clarify this issue?

6- Database and data collection: in this section researchers wrote “we excluded GI cancer patients who died before diagnosis of GI cancer”, might be a mistake in this sentence. Could you clarify this point?

7- Statistical analysis (and results): the authors performed univariate analysis but a multivariate analysis (not performed by authors), including most significant variables obtained from univariate (for example patient age), might strengthen results. Could you performed multivariate analysis? If not, add this in the study limitations.

Results:

8- This section should be reorganized: firstly describe your patient cohort.

For example in this section authors reported percentage of types of GI cancers included. “GI cancer included primarily colorectal cancer (n=514, 37.8%) and gastroesophageal cancer (n=502, 36.9%).

This sentence should be moved over, following “To evaluate the effect of statin use on patient outcomes, we studied 65,611 157 non-cancer patients with dyslipidemia and 1,361 dyslipidemia patients with GI cancer 158 (Table 1).” 

9- it might be interesting to know the stage of disease of GI cancer patients. Could you provide it? If not, add it among limitations.

10- Moreover authors wrote “These findings imply that statin use may be beneficial in dyslipidemia patients with GI cancer”. Please remember that the results are only descriptive, and enter any considerations into the Discussion section.

11- In the table 1 authors showed the association between continued statin use and mortality, but this table is not useful. Death is a time-to-event variable and not a categorical variable, therefore it should not be assessed by this kind of analysis.Instead of this, you could insert a table including patient characteristics. In the table 3 there is not the p value, could you add it please?

Discussion:

12- In this study authors demonstrated that GI cancer patients with higher MPR were associated with reduced mortality and that GI cancer patients with increased DDD/365 were associated with increased mortality. This data deserves a wider discussion.

Moreover, statin dose might be inaccurate and too generic.

In fact it is known that also statins are classified according to their intensity [Low-intensity (LDL-C reduction <30%), Moderate-intensity (LDL- C reduction 30% to <50%), High-intensity (LDL-C reduction >50%)], therefore, more than the dosage (too generic variable), it would be interesting to evaluate the impact of statin intensity on mortality.

Recently, several works considered statin intensity and results are interesting and requires further studies (Cantini L, Pecci F, Hurkmans DP, Belderbos RA, Lanese A, Copparoni C, Aerts S, Cornelissen R, Dumoulin DW, Fiordoliva I, Rinaldi S, Aerts JGJV, Berardi R. High-intensity statins are associated with improved clinical activity of PD-1 inhibitors in malignant pleural mesothelioma and advanced non-small cell lung cancer patients. Eur J Cancer. 2021 Feb;144:41-48. doi: 10.1016/j.ejca.2020.10.031. Epub 2020 Dec 14. PMID: 33326868.).

Do you have data available about statin intensity? You could add them to enrich your interesting study.

Reviewer 2 Report

a. average dose for every statins

b. histological stage of cancer at diagnosis

c. there are few references about this topic. Is it possible to add a few other references?

d. why percentage of statin treatment in dyslipidemic patients is so low in Korea?

e. how was calculated the MPR? it's not clear.

f. why only diabetic patients were excluded, and not also hypertensive or obese?

g how many drop-out for any statins and the reasons.

h. were there differences in outcomes based on LDL at the beginning or at the end of the study?

Reviewer 3 Report

This is a study that aims to assess post diagnostic statin use in patients with dyslipidemia free of cancer and with dyslipidemia and GI cancer. The study does not specify whether individuals initiate or are prevalent users at baseline (GI cancer diagnosis) hence it may be very susceptible to selection bias (for e.g. individuals on statins may have had myocardial infarction before etc - you do not detect benefit for cardiovascular?). Further, the exposure "quartile of statin use during follow-up" is defined from baseline = incorporates immortal time bias, which is a serious design flaw. Please see PMID 28822996 for a thorough discussion about these biases in relation to this subject. Given these methodological flaws and uncertainties I do not think these analyses provide any meaningful inference. Please redefine your study with appropriate causal inference methodology. Further table 1 should not be organized with those with different outcomes (death vs. survival). It should specify the different exposure groups (>25%, 50%, 75% and >75%) so that the readers can assess whether they are comparable in terms of demography, risk factors etc or if it is likely that selection bias is a major problem. For e.g. this presentation does not help me understand if the >75% group is categorized by less severe GI cancers with lower TNM score, better prognosis, more severe CVD etc, which I suspect.